# Effect of Phase Feeding, Space Allowance and Mixing on Productive Performance of Grower-Finisher Pigs

**DOI:** 10.3390/ani12030390

**Published:** 2022-02-07

**Authors:** Jordi Camp Montoro, Joana Pessoa, David Solà-Oriol, Ramon Muns, Josep Gasa, Edgar Garcia Manzanilla

**Affiliations:** 1Pig Development Department, Animal and Grassland Research and Innovation Centre, Teagasc, Moorepark, Fermoy, P61 C996 Co. Cork, Ireland; Joana.Pessoa@teagasc.ie (J.P.); egmanzanilla@gmail.com (E.G.M.); 2Animal Nutrition and Welfare Service, Department of Animal and Food Sciences, Universitat Autònoma de Barcelona, 08193 Bellaterra, Spain; David.Sola@uab.cat (D.S.-O.); Josep.Gasa@uab.cat (J.G.); 3Section of Herd Health and Animal Husbandry, School of Veterinary Medicine, University College Dublin, Belfield, D04 V1W8 Dublin, Ireland; 4M3-BIORES-Measure, Model & Manage Bioresponses, KU Leuven, B-3001 Leuven, Belgium; 5Agri-Food and Biosciences Institute, Large Park, Hillsborough, Co Down BT 26 6DR, Northern Ireland, UK; Ramon.Muns@afbini.gov.uk

**Keywords:** group size, growth, lysine, pig, regrouping, swine

## Abstract

**Simple Summary:**

Low space allowance and mixing may affect productive performance in pigs, but it is not clear whether space allowance and mixing interact with each other during the grower-finisher period. In addition, phase feeding is a good strategy used to reduce feed costs and improve feed efficiency, but it may result in nutrient limitations in suboptimal production conditions, e.g., high stress. This study investigated the effect of space allowance, mixing and phase feeding on productive performance of grower-finisher pigs. Three trials were conducted. Space allowance did not affect productive performance in any of the trials. Non-mixed pigs were 5.40 and 5.25 kg heavier than mixed pigs at 21 weeks of age in trial 1 and trial 2. Introducing a second diet with lower lysine/energy ratio from 0.95 to 0.82 g/MJ at 15–16 weeks of age reduced productive performance of pigs by 3.45 and 4.05 kg at 21 weeks of age in trial 1 and trial 2, but not in trial 3 where all pigs were mixed. In conclusion, mixing and reducing lysine/energy ratio from 0.95 to 0.82 g/MJ at 15–16 weeks of age, had a more marked impact on productive performance than a reduction in space allowance from 0.96 to 0.78 m^2^/pig.

**Abstract:**

This study investigates the effects of space allowance (SA), mixing and phase feeding (PF) on performance of grower-finisher pigs. Three trials (T) were conducted. In T1 and T2, 345 pigs/trial were moved to finisher stage at 11 weeks of age and assigned to two SAs: 0.96 (n = 15 pens; 10 pigs/pen) and 0.78 (n = 15; 13 pigs/pen) m^2^/pig. Mixing was applied to 5 pens of each SA leading to a 2 × 2 factorial arrangement (SA × Mixing). For PF, 2 diets with 0.95 and 0.82 g SID Lys/MJ NE were applied to 5 pens of each SA (not mixed) leading to another 2 × 2 factorial arrangement (SA × PF). In T3, 230 pigs were moved to the grower-finisher stage at 11 weeks of age, mixed, and assigned to 4 treatments (SA × PF; n = 5 pens). Data were analyzed using general linear mixed models. SA did not affect performance (*p* > 0.05). Non-mixed pigs were 5.40 (T1) and 5.25 (T2) kg heavier than mixed pigs at 21 weeks of age (*p* < 0.001). PF reduced performance of pigs by 3.45 (T1) and 4.05 (T2) kg at 21 weeks of age (*p* < 0.001). In conclusion, mixing and reducing SID Lys:NE ratio from 0.95 to 0.82 g/MJ at 15–16 weeks of age, have a more marked impact on performance than reducing SA from 0.96 to 0.78 m^2^/pig.

## 1. Introduction

Adjusting space allowance (SA), mixing and phase feeding (PF) are common strategies that have been used over the years in pig production to facilitate management and reduce production costs. Space allowance plays a critical role in pig production from a productive performance, animal welfare and economic standpoint [1,2,3]. Increasing the number of pigs per pen is correlated with a decrease in the facility cost per pig produced [2], which is the second most expensive cost after feed cost in the grower-finisher period [4]. However, increasing the number of pigs per pen decreases the SA and feeder space per animal and pig performance could be affected with a decrease in average daily gain (ADG), driven by a decrease on average daily feed intake (ADFI) [1,5,6]. Space allowance is normally expressed as m^2^/pig although it changes as pigs grow [7]. Then, SA in m^2^ can be expressed using the following formula: SA = *k* × BW^0.667^, where *k* is a coefficient and BW^0.667^ represents the geometric conversion of body weight (BW) in kg to area [8]. Previous research reported a critical *k*-value of 0.0336, below which productive performance is affected [7]. Nevertheless, recent studies reported that performance may be affected above the 0.0336 *k* value [1,6,9]. A minimum space per pig (m^2^/pig) is required by the EU legislation [10], which varies for each physiological phase and it must be of 0.65 m^2^/pig at least when pigs weigh between 85 to 110 kg of BW, in order to improve and guarantee a minimum animal welfare. With the current societal concerns regarding animal welfare, pig producers have to maximize production efficiency and minimize facility cost, while at the same time, meeting animal welfare and farm sustainability requirements. 

Mixing is a common management strategy used to sort pigs by weight in order to decrease BW variability and facilitate farm management practices. However, previous research stated that mixing fails to reduce final BW variability in individual pig weights within a pen [11,12], and that it may affect growth performance [13,14,15] and pig welfare due to the establishment of a new social hierarchy after re-grouping [16].

Phase feeding is another common management strategy used in pig production. Grower-finisher feeding programs normally include between one to four diets. NRC [17] recommends two growing and two finishing feeding phases. However, Garry et al. [18] found no differences in growth performance between using one diet or two to four diets in Irish pig farms. On the other hand, recent research pointed out that PF reduces the feed cost and improves growth performance as it matches the pig’s nutrient requirements that change as a consequence of age or size [19,20]. Consequently, feed efficiency is improved, which leads to better farm sustainability because of reduced ammonia emissions, as the requirements of crude protein (CP) are reduced by age [19,20,21]. Menegat et al. [22] stated that it is feasible to simplify PF up to two dietary phases considering the compensatory growth capability in grower-finisher pigs. The latter would maximize feed efficiency utilization and facilitate feed management. 

Previous studies reported findings on the effects of SA and PF on growth performance in grower-finisher pigs [1,4,22]. However, very little attention has been paid to the role of mixing on productive performance in grower-finisher pigs over the last two decades [13,14,15]. Moreover, there has been little discussion about how SA and mixing or whether SA and PF interact with each other in field conditions [13,15]. Knowing how these common management strategies affect productive performance, would guide farmers, veterinarians, and advisors to make better management decisions in pig production. 

We hypothesized that, for grower-finisher pigs, SA and mixing would have an effect on pig productive performance, and that phase feeding would result in a better feed efficiency and similar growth performance to non-phase feeding. Therefore, the present study aimed to investigate and quantify the effect of SA, mixing and phase feeding on productive performance in single wet-dry feeder pens during the grower-finisher stage.

## 2. Materials and Methods

### 2.1. Animals, Experimental Design and Diets 

Three trials were conducted at the Teagasc Pig Research Facility in Fermoy, Co. Cork, Ireland. All trials received ethical approval from the Teagasc Animal Ethics Committee (TAEC 204/2018). A schematic illustration of the experimental designs for trials 1, 2 and 3 is provided in Figure 1. 

Trials 1 and 2 had the same experimental design. Danish Duroc × (Large White × Landrace) pigs born within one week (Birth weight = 1.46 ± 0.28 kg; average litter size = 15.2 ± 4.0; 9.7% piglets with <1 kg birth weight) were followed per pen as intact litters from birth until 11 weeks of age. Pigs were weaned at approximately 28 days and received a starter diet (200.0 g CP, 12.3 MJ of net energy (NE) and 14.0 g Standardized ileal digestible lysine (SID Lys) per kg of feed) for seven days, link diet (190.0 g CP, 11.0 MJ/NE and 12.8 g SID Lys per kg of feed) for 18 days and soybean meal–barley–wheat based nursery diet (178.0 g CP, 10.6 MJ/NE and 10.4 g SID Lys per kg of feed) for 28 days. At 11 weeks of age, pigs were moved to the grower-finisher stage. Pigs were housed in pens with fully slatted concrete floor (2.4 × 4.2m) containing one wet-dry feeder (330 mm (Width) × 370 mm (Depth) × 1000 mm (Height); MA37, Verba, Netherlands) and one nipple drinker. Water and pelleted feed were provided ad libitum. Temperature was automatically controlled by a Big Dutchman 135pro ventilation controller (Vechta, Germany), with water heating, air intake via a perforated ceiling, and mechanical exhaustion of stale air via a fan. Temperature in the finisher accommodation ranged from 17 °C and 21 °C. Artificial lighting (LED) was provided at a minimum light intensity level of no less than 40 Lux and typically averaging 140–160 Lux for a minimum continuous period of eight hours per day. Lighting was provided between 07:00 a.m.–18:00 p.m. every day to coincide with natural day light. Pens were enriched with a 1.20 m fixed larch wood post on one of the walls without impairing on the available floor space. Pigs were fed a soybean meal-maize-wheat based grower-finisher diet (161.8 g CP, 9.67 MJ/NE and 9.2 g SID Lys per kg of feed; Table 1) in both trials.

When moved to the grower-finisher stage at 11 weeks of age, pigs (N = 345 pigs, 34.4 ± 3.96 kg of BW in Trial 1; N = 345 pigs, 31.2 ± 4.20 kg of BW in Trial 2) were weighed per pen and were randomly assigned to two different SAs balanced by BW: 0.96 m^2^/pig (n = 15 pens; 10 pigs/pen) and 0.78 m^2^/pig (n = 15 pens; 13 pigs/pen). Litter pens were adjusted to the SA treatments by removing pigs in case it was necessary, keeping the same average BW per pen and similar standard deviation, and balanced per sex. Both SAs were above the minimum space per pig (0.65 m^2^/pig) set by the European legislation based on live weight [10]. The 15 pens in each SA treatment were arranged as follows (Figure 1):Five pens were left as intact litters and fed a single finisher diet (0.95 g/MJ SID Lys:NE) up to slaughter.Five pens were mixed balancing by weight and sex and fed a single finisher diet (0.95 g/MJ SID Lys:NE) up to slaughter. This resulted in a 2 × 2 factorial arrangement with SA and mixing as main factors.Five pens were left as intact litters and PF was applied with a second diet (0.82 g/MJ SID Lys:NE) introduced at 15 weeks of age in trial 2 (50.6 ± 5.93 kg of BW) and at 16 weeks of age in trial 1 (66.2 ± 5.65 kg of BW). This resulted in a 2 × 2 factorial arrangement with SA and PF as main factors.

Table 1 shows the composition of the diets in both trials. Diets were formulated to meet or exceed the minimum requirements established by the standard nutritional tables [17]. Pigs remained in the facility until they reached at least 110 kg of BW when they were sent to slaughter. Recording of the performance data was terminated when the first group of pigs went to slaughter. 

In trial 3, Danish Duroc × (Large White × Landrace) grower-finisher pigs (N = 230) born within one week were moved to the grower-finisher stage at 11 weeks of age (32.9 ± 5.92 kg of BW), mixed, balanced by BW and sex, and randomly assigned to the same two SAs (0.78 and 0.96 m^2^/pig; n = 10 pens) as in trials 1 and 2. Pigs received the same management from birth until 11 weeks of age as in the previous trials. Mixing criteria and housing conditions were also the same as in the previous trials. Within each SA, pens were allocated to a single diet or PF, leading to a 2 × 2 factorial arrangement considering SA and PF. The diets used in trial 3 were the same as in trials 1 and 2 (Table 1). The second diet in the case of PF was introduced at 15 weeks of age (53.7 ± 9.02 kg of BW). Pigs remained in the facility until they reached at least 110 kg of BW when they were sent to slaughter. Recording of the performance data was terminated when the first group of pigs went to slaughter.

In all trials, SA coefficient (*k*) for each treatment were calculated using the formula SA = *k* × BW^0.667^ [8]. The *k*-values are reported in Table 2.

### 2.2. Measurements

#### 2.2.1. Body Weight, Feed Intake and Feed Efficiency Traits 

Pigs were weighed per pen every two weeks until pigs started to go to slaughter at 21 weeks of age when the first pigs reached the target slaughter weight, 110 kg. Feed intake was recorded daily at pen level. Average daily gain, ADFI and feed conversion ratio (FCR; $\frac{kg\;of\;feed\;consumed}{\;kg\;of\;BW\;gain}$) were calculated for every 2 weeks interval.

#### 2.2.2. Carcass Traits 

Cold carcass weight (CCW), fat thickness (FT), muscle content (MC) and percentage of lean meat (LM%) were recorded by the slaughterhouse personnel. Cold carcass weight, FT and MC were measured immediately after the processing line. Fat thickness and MC were measured using a Fat-O-Meat’er (Carometec Food Technology, Carometec A/S, Hasselunden 9, Smørum, Denmark). Percentage of lean meat was calculated according to the formula established by the European Communities Pig Carcass Grading Amendment Regulations [23]:$$\%\;\mathrm{lean}\;\mathrm{meat}\;=60.30-\left(0.847\times \mathrm{fat}\;\mathrm{thickness}\right)+\left(0.147\times \mathrm{muscle}\right)$$

### 2.3. Statistical Analyses 

All data were analyzed using SAS v9.4 (SAS Institute Inc., Cary, NC, USA). In trial 1 and trial 2, the model included SA, PF or mixing, and their interaction as fixed effects. In trial 3, the model included SA, PF, and their interaction as fixed effects. Initial BW was used as a co-variable for BW, ADG, ADFI and FCR, and each pen was considered as the experimental unit. Data were analyzed using general linear mixed models, accounting for repeated measurements in all trials. In terms of carcass traits, predicted variables included LM%, FT and MC, and each pen was considered as the experimental unit. In trial 1 and trial 2, model included age at slaughter, CCW, SA, PF or mixing, and the SA × PF or mixing interaction as fixed effects. In trial 3, model included age at slaughter, CCW, SA, PF, and the SA × PF interaction as fixed effects. Data were analyzed using general linear models. Multiple means comparisons were carried out using Tukey-Kramer’s correction. Alpha level for determination of significance was 0.05, and from 0.05 to 0.10 for trends. Results for fixed effects are reported as least square means ± standard error. 

## 3. Results

### 3.1. Body Weight, Feed Intake and Feed Efficiency Traits 

Only general results of the whole trial period for each productive performance variable are presented. 

#### 3.1.1. Space Allowance × Mixing (Trials 1 and 2) 

Body weight showed an interaction between SA and mixing in trial 1. Non-mixed pigs with 0.78 m^2^/pig SA were 6.1 and 6.5 kg heavier than mixed pigs with 0.96 and 0.78 m^2^/pig SA, respectively, at 21 weeks of age (*p* < 0.001; Table 3). On average, non-mixed pigs gained 74 g more per day (*p* = 0.004) and consumed 101.8 g more of feed per day (*p* = 0.007) than mixed pigs for the whole period of the trial (Table 3). Non-mixed pigs tended to have lower FCR compared to mixed pigs from 11 to 21 weeks of age (*p* = 0.079; Table 3). 

In trial 2, an interaction between SA and mixing on BW was observed (*p* = 0.005), but was not significant after adjustment for multiple mean comparison (*p* > 0.05; Table 3). Non-mixed pigs were 5.25 kg heavier than mixed pigs at 21 weeks of age (*p* < 0.001; Table 3). On average, non-mixed pigs gained 76.6 g more per day (*p* = 0.029) and consumed 144.4 g more of feed per day (*p* = 0.046) than mixed pigs for the whole period of the trial (Table 3). Pigs with 0.78 m^2^/pig SA had lower FCR compared to pigs with 0.96 m^2^/pig SA for the whole period of the trial (*p* < 0.001; Table 3). 

In both trials, the mixing effect was observed as a long-lasting effect, with the difference between mixed and non-mixed increasing towards the end of the grower-finisher period, and not at the beginning when the pigs were mixed (Figure 2).

#### 3.1.2. Space Allowance × Phase Feeding (Trials 1, 2 and 3) 

Body weight showed an interaction between SA and PF in trial 1. Pigs with 0.78 m^2^/pig SA and 0.95 g/MJ SID Lys:NE diet were 4.4 and 4.5 kg heavier than pigs with 0.82 g/MJ SID Lys:NE diet and 0.78 or 0.96 m^2^/pig SA, respectively, at 21 weeks of age (*p* < 0.001; Table 4). On average, pigs that received the 0.95 g/MJ SID Lys:NE diet gained 90.1 g more per day (*p* = 0.011) and had lower FCR (*p* = 0.005) compared to pigs that received the 0.82 g/MJ SID Lys:NE diet from 16 to 21 weeks of age (Table 4). No differences were observed on ADFI between treatments (*p* > 0.05; Table 4). 

In trial 2, an interaction between SA and PF on BW was observed. Pigs with 0.96 m^2^/pig SA and 0.95 g/MJ SID Lys:NE diet were 5.6 and 5.2 kg heavier than pigs with 0.82 g/MJ SID Lys:NE diet and 0.78 or 0.96 m^2^/pig SA, respectively, at 21 weeks of age (*p* < 0.001; Table 4). On average, pigs that received the 0.95 g/MJ SID Lys:NE diet gained 83.9 g more per day (*p* = 0.013) and had lower FCR (*p* = 0.031) compared to pigs that received the 0.82 g/MJ SID Lys:NE diet from 15 to 21 weeks of age (Table 4). No differences were observed on ADFI between treatments (*p* > 0.05; Table 4). 

In trial 3, BW showed an interaction tendency between SA and PF at 21 weeks of age (*p* = 0.093; Table 4). Average daily gain, ADFI and FCR results for the whole trial period were not affected by SA and PF (*p* > 0.05; Table 4).

### 3.2. Carcass Traits

#### 3.2.1. Space Allowance × Mixing 

Lean meat %, FT and MC were not affected by SA and mixing on both trial 1 and trial 2 (*p* > 0.05; Table 5). 

#### 3.2.2. Space Allowance × Phase feeding 

Lean meat % and FT were not affected by SA and PF in trial 1, trial 2 and trial 3 (*p* > 0.05; Table 6). Muscle content was not affected on both trial 1 and trial 2 (*p* > 0.05; Table 6). However, MC showed an interaction between SA and PF in trial 3 (*p* = 0.010), although no significant difference was observed in the least square means (*p* > 0.05; Table 6). 

## 4. Discussion

### 4.1. Body Weight, Feed Intake and Feed Efficiency Traits

This study was carried out to assess the importance of space allowance, mixing and phase feeding on productive performance of grower-finisher pigs. A pen system of 10–13 grower-finisher pigs linked to one wet-dry feeder, which optimizes feed efficiency [24] and allows a good assessment of health and welfare issues in pigs, was chosen because it is one of the most common systems in growing-finishing units in the EU. The latter it is suggested by the authors’ experience in groups such as ECPHM or EUPIG (https://www.eupig.co.uk, accessed on 12 December 2021). Moreover, space allowances in the present study were not chosen to ensure a negative effect on productive performance and were chosen above the 0.65 m^2^/pig minimum set by the current EU legislation based on live weight [10]. This decision was taken as a consequence that a space allowance of 0.65 m^2^/pig is already considered negative from a welfare point of view because of a very low amount of shared space [25], and its final *k*-value is below the critical *k*-value when pigs reach the market slaughter weight, which was 110 kg of BW per commercial practice. Furthermore, under the conditions of the study, space allowances were adjusted by changing the number of grower-finisher pigs per pen. This method ensures a feasibility at producer level because it is how would happen in field conditions, although it can induce confounding of space allowance with group size [12]. Nevertheless, previous studies have suggested that group size does not have a detrimental effect on growth performance of grower-finisher pigs [26,27], and it is not a predictor for ADG, ADFI and FCR [4]. Moreover, a higher number of pigs per pen decreases the feeder space, which could have also affected productive performance [28]. However, Wastell et al. [9] observed that feeder space did not have a detrimental effect on growth performance between 10 and 13 pigs per wet-dry feeder space in a space allowance of 0.78 m^2^/pig. Together, these studies suggest that group size and feeder space would not affect the productive performance of grower-finisher pigs under the conditions of this study. Therefore, and regarding the pen system studied, the present study discusses space allowance as the main factor to affect productive performance, instead of group size or feeder space.

Space allowance did not affect final BW and overall ADG and ADFI, which is in line with results obtained from a previous study using similar space allowances [15]. However, previous literature suggested that further decreasing space allowances results in poorer productive performance [1,5,29]. These studies subjected pigs to restricted space allowances below 0.70 m^2^/pig, which may explain the difference in results. Moreover, other studies found that 0.80 m^2^/pig space allowance affected productive performance when pigs were marketed to at least 138 kg of target slaughter BW [30]. Overall, all these studies could be compared using the allometric approach which can relate space allowance to body weight [8]. Gonyou et al. [7], using a broken-line analysis, reported a critical *k*-value of 0.0336, below which productive performance was affected. This *k*-value was almost reached in pigs with 0.78 m^2^/pig by the end of the trial when they reached the target slaughter weight (i.e., 110 kg). Then, productive performance may not be affected during the grower-finisher period when space allowance is established based on the critical *k*-value at the target slaughter weight. Nevertheless, recent studies reported that productive performance restriction could occur above the critical *k*-value of 0.0336 [1,31], which may depend on the genetics and feeder system [9]. Moreover, it has been suggested that the critical *k*-value could also underestimate the space allowance requirement for marketed heavy pigs with body weights over 120 kg [30]. Nonetheless, body weights over 120 kg were not reached in the present study. 

Pigs in the lower space allowance had a better feed efficiency in trial 2, although this finding is not consistent in the study as it was not observed in trial 1. Similarly, there are several conflicting reports in previous literature of FCR not being affected by space allowance [1,6,7], being increased by decreasing space allowance [5,14,31], or being improved with lower space allowances [9]. These discrepancies within the literature could rely on the experimental design itself of each prior study and which space allowances were considered low or high. Moreover, environmental and management factors such mixing or the farm health status may have their role in feed efficiency [15,32]. 

Mixing appears to have a considerable effect on productive performance in grower-finisher pigs. This finding is in accordance with previous studies which found reduced performance when pigs were mixed at the beginning of the grower-finisher period and followed in a 4-wk experiment with space allowances of 0.25 and 0.52 m^2^/pig [13,14] or when pigs had 83 kg of BW and were mixed and followed in a 2-wk experiment [33]. Thus, the present study suggests that the strategy of mixing should be reconsidered in current modern facilities and genetics. Moreover, the issue of mixing segregating by sex, which it is a common practice, is an intriguing one which could be usefully explored in further research. Apart of the detrimental effect on productive performance [15], mixing affects animal welfare [34] and sorting by weight fails to improve final variability in individual pig weights within a pen [11], although it may facilitate farm management. Therefore, strategies to avoid or mitigate mixing, such as maintaining intact litters, should be considered in future research. The management strategy of keeping pigs in intact litters is possible in farrow-to-finish farms, but it may be more difficult in farms with the 3 production stages separated, where animals need to be transported using a truck.

Contrary to expectations, mixing was found to have a long-term effect instead of an initial effect on growth performance associated with aggression. Mixed pigs had a poorer growth performance during the second period of the grower-finisher stage. This might be explained due to chronic stress related to aggression and social hierarchy [16,35,36]. Chronic stress can lead to immunosuppression and could, ultimately, have detrimental implications for pig health and performance [35,36,37]. Recently, Foister et al. [16] suggested that high betweenness centralization after mixing represents poorly established dominance relationships between pen mates. This leads to chronic stress due to prolonged aggression behavior. Nonetheless, several studies have reported that mixing does not have an effect on health and performance traits when it is applied to larger group sizes of 20, 40, or 80 pigs [26], or up to 108 pigs [27]. Turner et al. [38] observed that mixed pigs in large group size (i.e., 80 pigs) displayed a marked reduction in aggressive behavior towards foreign pigs, perhaps due to different social dynamics in larger groups. The latter may explain the differences between the present study when there are only 10 or 13 pigs per group and the previous studies with higher group sizes.

Although this study aimed to find whether space allowance and mixing interacted, the interaction observed between space allowance and mixing in final BW in trials 1 and 2 were contradictory in terms of space allowance. Moreover, the interaction did not show up in the ADG, ADFI or FCR, and it could be checked for repeatability in further experiments. Previous research suggested that the effects of reduced space allowance and mixing were additive [13]. 

Reducing the SID Lys:NE ratio from 0.95 to 0.82 g/MJ at 15–16 weeks of age had a detrimental effect on growth performance in trials 1 and 2. This outcome is contrary to previous studies, which have reported that a single phase feeding strategy reduces overall performance [22] or has no impact on growth performance compared to multiple dietary phases [18,39,40]. Nevertheless, the divergence between previous research and the present study may rely on differences in the SID Lys:NE ratio, weight ranges and the experimental conditions. Previous research considered compensatory growth during the grower-finisher stage and provided low SID Lys:NE ratio initially and adequate or excess SID Lys:NE ratio later in the grower-finisher period [18,22,39]. However, this was not the case in the present study, as pigs received an adequate SID Lys:NE ratio (i.e., 0.95 g/MJ) at the beginning of the trial and the 0.82 g/MJ SID Lys:NE ratio applied as phase feeding strategy at 15–16 weeks of age was adjusted to meet or exceed the requirements established by the standard nutritional tables [17]. Therefore, it is somewhat surprising that productive performance was affected using this phase feeding strategy. A possible explanation for these results might be related to the genetic advancements, where high lean gain potential pig genetics have higher SID Lys requirements [41,42,43]. Nowadays, the Lys requirements for those high lean gain potential pig genetics are already higher than the NRC recommendations [42,43,44], and it should be considered to what extent current recommendations are aligned with the current genetic pig potential. Nonetheless, different genotypes may derive to different results. Thus, future studies could consider predicting dietary specifications using the NRC model by estimating the lean gain potential of the pigs being studied. Another possible explanation for the results observed in the present study may be due to large changes in the SID Lys:NE ratio, which may not be favorable to maximize grower-finisher pig productive performance [45]. Moreover, environmental factors such as management, feedstuffs, facilities or the farm health status may have an effect on pig’s nutrient requirements [46,47,48]. Management reasons could explain that no differences in productive performance were observed between pigs that were fed one-single diet and pigs that had the phase feeding in trial 3 where all pigs were mixed. Mixing effect on productive performance was even more severe than reducing SID Lys:NE from 0.95 to 0.82 g/MJ in trial 1 and 2. Mixing could have had an impact on productive performance in both groups that may have concealed the phase feeding effect in trial 3. Likewise, the interaction observed between space allowance and phase feeding in final BW in both trials 1 and 2 was not evident when all pigs were mixed in trial 3. Therefore, the relationship between space allowance, phase feeding and productive performance may vary between batches, due to interactions of different environmental/management factors such as mixing and should be checked for repeatability in further studies. Finally, it is worth mentioning that the high levels of SID Lys:NE ratio used at the late finishing stage may increase the levels of NH_3_ [47,49,50] which is contrary to farm sustainability because it contributes to the environment pollution [21]. 

### 4.2. Carcass Traits

Space allowance × mixing and space allowance × phase feeding were not a source of variation for carcass traits. These results match those observed in earlier studies which did not find differences in carcass traits using different phase feeding strategies [18,22,40]. Nevertheless, previous literature also found that lean meat % increases as space allowance decreases [1], and fat thickness increases as SID Lys:NE ratio decreases [48,49,51]. Although these findings were not observed in the present study, an interaction tendency was observed between space allowance and phase feeding on lean meat % and fat thickness in trial 1, and these two traits were numerically different in trial 2 when reducing the SID lys:NE ratio from 0.95 to 0.82. Therefore, further work is needed to fully understand the implications of space allowance, phase feeding and mixing on carcass traits.

## 5. Conclusions

The present study compared the effects of space allowance, mixing and phase feeding during the grower-finisher period in pens of 10–13 grower-finisher pigs with one single wet-dry feeder. Space allowance of 0.78 m^2^/pig did not affect productive performance compared to 0.96 m^2^/pig. Space allowance may not compromise growth performance if the critical *k*-value is considered at the marketed weight. Mixing causes a negative and long-term impact on productive performance. Reducing SID Lys:NE ratio from 0.95 to 0.82 g/MJ at 15–16 weeks of age had an effect on productive performance when pigs were not mixed. Nevertheless, carcass traits were not affected as a consequence of mixing or reducing SID Lys:NE ratio from 0.95 to 0.82 g/MJ at 15–16 weeks of age. Finally, high lean gain potential pig genetics may require a higher SID Lys:NE ratio than current nutrient requirement recommendations during the grower-finisher period.

## Figures and Tables

**Figure 1 animals-12-00390-f001:**
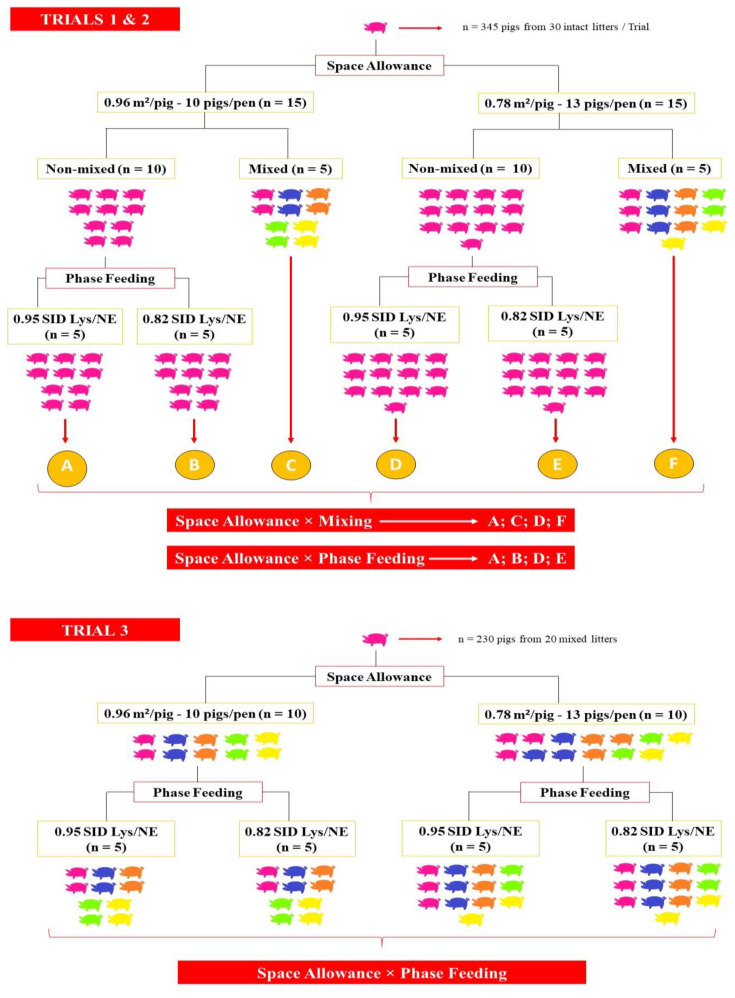
Schematic illustration of the experimental design in trial 1, 2 and 3. In trials 1 and 2, experimental design was a two 2 × 2 factorial arrangement with space allowance and mixing as treatments and space allowance and phase feeding as treatments. In trial 3, experimental design was a 2 × 2 factorial arrangement with space allowance and phase feeding as treatments, and mixing was applied to all the pens. Space allowances were above the minimum space per pig set by European legislation based on live weight [10].

**Figure 2 animals-12-00390-f002:**
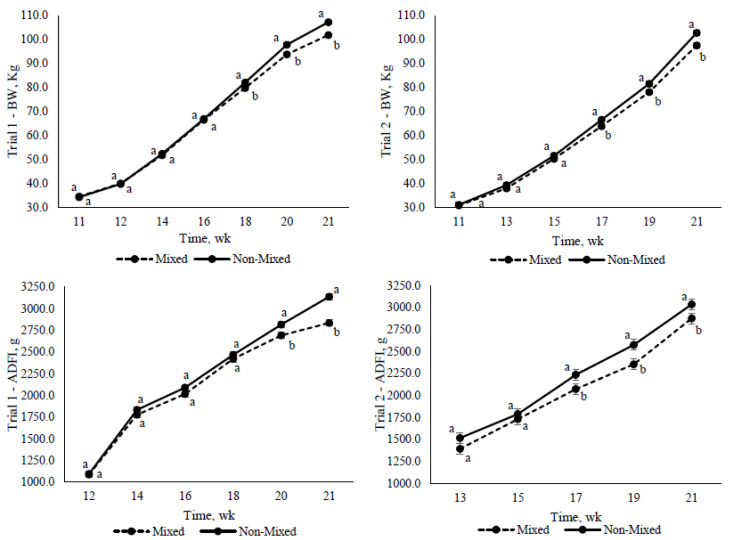
Effect of mixing on body weight (BW) and average daily feed intake (ADFI) in trial 1 and trial 2 (n = 5) from 11 to 21 weeks of age, when the first group of pigs reached 110 kg of BW and were sent to slaughter; ^a,b^ Significant differences between groups (*p* < 0.05).

**Table 1 animals-12-00390-t001:** Composition on an as-fed basis of the experimental diets.

	Diets	
	High SID Lys:NE Ratio ^1^	Low SID Lys:NE Ratio ^2^
**Ingredients, g/kg**		
Wheat	435.0	435.0
Corn	300.0	351.0
Soybean meal 48	171.0	120.0
Soybean hulls	71.0	71.0
Calcium carbonate	11.0	11.0
Dicalcium phosphate anhydrous	1.00	1.00
Sodium chloride	3.00	3.00
L-Lysine HCl	3.75	3.75
L-Threonine	1.70	1.70
DL-Methionine	0.93	0.93
L-Tryptophan	0.15	0.15
Vitamin and trace mineral mixture ^3^	1.47	1.47
**Calculated Composition ^4^, % as fed or as specified**		
Dry Matter	87.34	87.26
NE, MJ/kg	9.67	9.83
Crude Protein	16.18	14.28
Total Lys	1.03	0.90
SID Lys	0.92	0.80
SID Thr	0.64	0.58
SID Met	0.31	0.29
SID Trp	0.17	0.15
Fat	2.23	2.33
Crude Fiber	5.06	4.87
NDF	14.60	14.51
Calcium	0.58	0.57
Digestible Phosphorus	0.23	0.22
SID Lys:NE, g/MJ	0.95	0.82

^1^ Higher ratio SID Lys diet was given to the pigs all the way or during 11 to 15–16 weeks of age to the phase feeding treatment group. ^2^ Lower ratio SID Lys diet was given to the phase feeding treatment group from 15–16 weeks of age until slaughter. ^3^ Provided per each kg of feed: 60 mg Copper sulphate, 80 mg Ferrous sulphate monohydrate, 50 mg Manganese oxide, 100 mg Zinc oxide, 0.5 mg Potassium iodate, 0.4 mg Sodium selenite, 2 MIU Vitamin A, 0.5 MIU Vitamin D3, 40 MIU Vitamin E, 4 mg Vitamin K, 0.015 mg Vitamin B12, 2 mg Riboflavin, 12 mg Nicotinic acid, 10 mg Pantothenic acid, 2 mg Vitamin B1, 3 mg Vitamin B6. ^4^ The calculated composition is based on reference data provided by INRAE. NE = Net Energy; SID = Standardized Ileal Digestible; NDF = Neutral Detergent Fiber.

**Table 2 animals-12-00390-t002:** Initial and final space allowance coefficient (*k*) for each treatment in trial 1, 2 and 3.

	Trial 1		Trial 2		Trial 3	
Number of pigs/pen	10	13	10	13	10	13
Space allowance, m^2^/pig	0.96	0.78	0.96	0.78	0.96	0.78
Space allowance coefficient, *k* ^1^						
Initial ^2^	0.090	0.074	0.098	0.078	0.093	0.077
Final ^3^	0.043	0.035	0.044	0.036	0.044	0.036

^1^ The allometric expression of the space coefficient is as follows: *k* = Space allowance (m^2^) / BW^0.667^ (kg). ^2^ Initial body weight in trial 1 was: 34.6 ± 3.76 kg (0.96 m^2^/pig), 34.1 ± 4.12 kg (0.78 m^2^/pig). In trial 2 was: 30.8 ± 4.00 kg (0.96 m^2^/pig) and 31.5 ± 4.36 kg (0.78 m^2^/pig). In trial 3 was: 33.3 ± 6.00 kg (0.96 m^2^/pig) and 32.5 ± 5.84 kg (0.78 m^2^/pig). ^3^ Final space allowance coefficient was predicted based on the average pen weight when the first group of pigs reached 110 kg of BW and were sent to slaughter.

**Table 3 animals-12-00390-t003:** Effect of space allowance and mixing on bodyweight (BW), average daily gain (ADG), average daily feed intake (ADFI) and feed conversion ratio (FCR) (Least square means (LS means) ± Standard error mean (SEM)) in trial 1 and trial 2 from 11 to 21 weeks of age.

	Space Allowance							
	0.96 m^2^/Pig		0.78 m^2^/Pig			*p*-Value		
Trait	Mixed (n = 5)	Non-Mixed (n = 5)	Mixed (n = 5)	Non-Mixed (n = 5)	SEM	Mixing	Space Allowance	Interaction
Trial 1								
BW ^1^, kg, 21 wk	102.1 ^b^	106.4 ^ab^	101.7 ^b^	108.2 ^a^	0.95	<0.001	0.472	<0.001
ADG, g	983.0	1034.1	955.4	1052.3	21.82	0.004	0.836	0.309
ADFI, g	2150.3	2222.3	2125.6	2257.1	32.57	0.007	0.880	0.374
FCR	2.18	2.12	2.19	2.11	0.04	0.079	0.974	0.692
Trial 2								
BW, kg, 21 wk	97.5	104.3	97.8	101.5	1.50	<0.001	0.400	0.005
ADG, g	914.5	1014.3	929.7	983.1	31.90	0.029	0.809	0.476
ADFI, g	2120.7	2314.7	2055.1	2149.9	66.80	0.046	0.110	0.468
FCR	2.36	2.32	2.23	2.18	0.03	0.172	<0.001	0.825

^1^ Initial body weight in trial 1 was: 34.2 ± 3.25 kg (*p* = 0.545). In trial 2 was: 30.8 ± 4.31 kg (*p* = 0.645). ^a,b^ Within rows, significant differences between groups (*p* < 0.05).

**Table 4 animals-12-00390-t004:** Effect of space allowance and phase feeding on bodyweight (BW), average daily gain (ADG), average daily feed intake (ADFI) and feed conversion ratio (FCR) (Least square means (LS means) ± Standard error mean (SEM)) in trial 1 from 16 to 21 weeks of age, and in trial 2 and trial 3 from 15 to 21 weeks of age^1^.

	Space Allowance							
	0.96 m^2^/Pig		0.78 m^2^/Pig			*p*-Value		
Trait	0.82 SID Lys:NE (n = 5)	0.95 SID Lys:NE (n = 5)	0.82 SID Lys:NE (n = 5)	0.95 SID Lys:NE (n = 5)	SEM	Phase feeding	Space Allowance	Interaction
Trial 1								
BW^2^, kg, 21 wk	103.2 ^b^	105.7 ^ab^	103.3 ^b^	107.7 ^a^	0.84	<0.001	0.208	<0.001
ADG, g	1084.7	1157.1	1094.3	1202.0	30.89	0.011	0.391	0.575
ADFI, g	2754.9	2748.0	2758.1	2839.1	42.19	0.398	0.280	0.312
FCR	2.56	2.39	2.54	2.37	0.05	0.005	0.662	0.943
Trial 2								
BW, kg, 21 wk	97.6 ^b^	102.8 ^a^	97.2 ^b^	100.1 ^ab^	0.92	<0.001	0.094	<0.001
ADG, g	1064.6	1174.7	1063.0	1120.4	29.59	0.013	0.361	0.386
ADFI, g	2524.1	2665.7	2510.0	2505.6	59.40	0.269	0.164	0.236
FCR	2.37	2.27	2.37	2.24	0.05	0.031	0.770	0.760
Trial 3								
BW, kg, 21 wk	100.7	101.0	98.0	100.6	0.93	0.120	0.095	0.093
ADG, g	1098.2	1102.5	1036.7	1094.0	21.20	0.164	0.118	0.230
ADFI, g	2521.6	2530.0	2409.9	2535.1	54.83	0.242	0.347	0.304
FCR	2.29	2.28	2.32	2.31	0.03	0.773	0.479	0.988

^1^ In trial 1 and trial 2 pigs were kept in litters, while in trial 3 pigs were mixed. ^2^ Initial body weight in trial 1 was: 33.6 ± 4.61 kg (*p* = 0.718). In trial 2 was: 30.7 ± 4.30 kg (*p* = 0.610). In trial 3 was: 32.4 ± 5.93 kg (*p* = 0.966). ^a,b^ Within rows, significant differences between groups (*p* < 0.05).

**Table 5 animals-12-00390-t005:** Effect of space allowance and mixing on lean meat %, fat thickness and muscle content (Least square means (LS means) ± Standard error mean (SEM)) from trial 1 and 2 ^1^.

	Space Allowance							
	0.96 m^2^/Pig		0.78 m^2^/Pig			*p*-Value		
Trait	Mixed (n = 5)	Non-Mixed (n = 5)	Mixed (n = 5)	Non-Mixed (n = 5)	SEM	Mixing	Space Allowance	Interaction
Trial 1								
Lean meat, %	57.71	57.98	58.07	57.60	0.273	0.741	0.984	0.173
Fat thickness, %	13.57	13.56	13.31	14.03	0.358	0.390	0.784	0.297
Muscle content, %	45.93	48.61	47.60	48.35	0.746	0.059	0.364	0.187
Trial 2								
Lean meat, %	58.62	58.49	58.52	58.77	0.342	0.853	0.824	0.565
Fat thickness, %	12.55	12.70	12.65	12.46	0.450	0.965	0.895	0.705
Muscle content, %	47.29	47.21	47.07	48.29	0.773	0.454	0.629	0.394

^1^ Model included age at slaughter, cold carcass weight, space allowance, mixing and their interaction as fixed effects.

**Table 6 animals-12-00390-t006:** Effect of space allowance and phase feeding on lean meat %, fat thickness and muscle content (Least square means (LS means) ± Standard error mean (SEM)) from trial 1, 2 and 3 ^1^.

	Space Allowance							
	0.96 m^2^/Pig		0.78 m^2^/Pig			*p*-Value		
Trait	0.82 SID Lys:NE (n = 5)	0.95 SID Lys:NE (n = 5)	0.82 SID Lys:NE (n = 5)	0.95 SID Lys:NE (n = 5)	SEM	Phase Feeding	Space Allowance	Interaction
Trial 1								
Lean Meat, %	57.16	58.03	57.90	57.77	0.293	0.230	0.426	0.106
Fat thickness, %	14.37	13.46	13.24	13.92	0.397	0.789	0.410	0.065
Muscle content, %	47.37	48.19	45.97	47.64	0.681	0.100	0.167	0.545
Trial 2								
Lean Meat, %	58.08	58.62	57.83	58.87	0.451	0.121	0.996	0.588
Fat thickness, %	13.17	12.53	13.65	12.18	0.614	0.131	0.920	0.512
Muscle content, %	46.48	47.18	47.65	47.06	0.826	0.952	0.535	0.444
Trial 3								
Lean Meat, %	57.80	58.49	58.02	58.07	0.220	0.112	0.662	0.161
Fat thickness, %	13.49	12.86	13.50	13.19	0.302	0.140	0.605	0.600
Muscle content, %	46.06	48.45	48.53	46.55	0.744	0.785	0.718	0.010

^1^ In trial 1 and trial 2 pigs were kept in litters, while in trial 3 pigs were mixed. Model included age at slaughter, cold carcass weight, space allowance, phase feeding and their interaction as fixed effects.

## Data Availability

The datasets used and analyzed during the current study are available from the corresponding author on reasonable request.

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
