# Peer review of "Effect of Phase Feeding, Space Allowance and Mixing on Productive Performance of Grower-Finisher Pigs"

_animals, 2022, doi:10.3390/ani12030390_

Round 1

Reviewer 1 Report

Dear authors,

The manuscript "Effect of phase feeding space allowance and mixing on productive performance of grower-finisher pigs" presents relevant information from the pig production improvement. Although there are some aspects that I would like to clarify:

  1. The sex of experimental animals in the trials.
  2. The practive of keeping the animals together in the same litter until week 11, do you think is possible on a commercial farm? If the presumption is negative, please specify this fact in the manuscript.
  3. The possibility of working in a mixed or non-mixed system depends both on the imnune setatus of the farm (and other aspects that you mentioned) as well as on the range of birth weights of the piglets from published studies. It would be very interesting to detail in the material and methods the live weight of the pigs at birth with its sd in all the  experimental groups, as well as to specify the percentage of pigs under 1 kg included in the study.
  4. Regarding the space allowance, I would like ask for the inclusion of temperature data recorded in the different experiments.

Thank you in advance!

Author Response

Thank you for your comments and revision. We clarified the points that you commented in the manuscript:

  1. The sex of experimental animals in the trials.

Author: Pens were balanced for sex in the mixed pens to match the litter pens. We clarified that point in the materials and methods section in lines 129, 134, and 148.

  1. The practice of keeping the animals together in the same litter until week 11, do you think is possible on a commercial farm? If the presumption is negative, please specify this fact in the manuscript.

Author: The practice of keeping pigs together in the same litter is possible in farrow-to-finish farms and we work with some commercial farms that do it in Ireland, but it may be more difficult in farms with the 3 separated production stages where animals need to be transported using a truck. We commented this fact in the discussion section in lines 364-366.

  1. The possibility of working in a mixed or non-mixed system depends both on the immune status of the farm (and other aspects that you mentioned) as well as on the range of birth weights of the piglets from published studies. It would be very interesting to detail in the material and methods the live weight of the pigs at birth with its sd in all the experimental groups, as well as to specify the percentage of pigs under 1 kg included in the study.

Author: This is a very relevant comment and an aspect that we had actually missed in the description. Thanks to the reviewer for spotting it. It is indeed very important that working with intact litters may not be possible when high numbers of piglets present low weights (<1kg). The birth weights by experimental group are not available for the present study however, we have added the average and SD weight of the batch, the percentage of animals below 1kg and the average litter size which may be relevant when extrapolating these results for other situations.  

  1. Regarding the space allowance, I would like ask for the inclusion of temperature data recorded in the different experiments.

Author: We have clarified the average temperature in the finisher rooms where the study was conducted in lines 113-116.

Reviewer 2 Report

The manuscript submitted for review presents the effect of space allowance, mixing and phase feeding on productive performance in single wet-dry feeder pens during the grower-finisher stage.The experiment is well-designed and allowed for the intended results. The results are presented in a very clear way and their discussion allows their application in breeding practice. I encourage the Editors to publish it, after taking into account some of my suggestions:
Line 28: abbreviation (SA) should be explained before it is used for the first time
Line 125: I suggest moving the Figure A1 closer to its discussion to make it easier to read.
Line 118: How pigs were allotted to pens?Was it randomly allotation?

Author Response

Thank you for your comments and revision. We have addressed your suggestions in the manuscript.

  1. Line 28: abbreviation (SA) should be explained before it is used for the first time.

 Author: We are sorry for that error. We have changed “SA” for “space allowance” in line 28.

  1. Line 125: I suggest moving the Figure A1 closer to its discussion to make it easier to read.

Author: We have addressed your suggestion and moved Figure A1 closer to its discussion.

  1. Line 118: How pigs were allotted to pens? Was it randomly allotation?

Author: We have addressed your comment and corrected it in lines 126 and 148. 

Reviewer 3 Report

I recommend addressing the following improvements. In addition English writing must be improved by correcting it by a native English speaker, and typos removed. Try the best to improve the quality and style of the manuscript writing.

The graphical presentation of the methodology makes it easier to understand the written text but after a long reflection. Please consider rewriting the section materials and methods

  1. Most of the rations are composed and recommended for specific phases of fattening, based on the maintenance in optimal environmental conditions. As a result, during fattening in excellent conditions, the animals will consume more feed and spend less of its value on growth because they use the feed ingredients to heat the body. On the other hand, in hot conditions, feed intake is significantly reduced, and it becomes impossible to provide the right amount of nutrients. The publication only mentions temperature, how were the animals held? Please insert text about husbandry conditions: lighting intensity and duration, gaseous ammonia, temperature, etc.
  2. Why were specifically the described genotypes chosen. Please discuss.(Different genotypes may also need different fatting lengths.)
  3. General remarks:
  • No commas in lines: 45; 46; 48; 51; 55; 66 ….
  • Line 52: in average, not on average
  • Line 73: (…) However, Garry et al. [18] found no differences on in growth performance between using one single diet or two to four diets in Irish pig farms. (…)
  • Line 77: as a consequence -> due to
  • Line 78: (…)“a better”(…)
  • Lines 92-95: (…)Therefore, the aim of the present study was aimed to investigate and quantify the effect of space allowance, mixing (…)
  • Line: 283: (…)This study was carried out with the aim of assessing to assess the importance of space allowance, mixing, and phase feeding on the productive performance of grower-finisher pigs (…)
  • Line 361: (…)Although this study aimed to find whether space allowance and mixing interacted with each other, the interaction observed between space allowance and mixing in final BW in both trials 1 and 2 was were contradictory in terms of space allowance.(…)

Author Response

Thank you for your comments and revision. The English of the manuscript has been reviewed by a native English speaker as recommended. We clarified the comments in the manuscript:

  1. Most of the rations are composed and recommended for specific phases of fattening, based on the maintenance in optimal environmental conditions. As a result, during fattening in excellent conditions, the animals will consume more feed and spend less of its value on growth because they use the feed ingredients to heat the body. On the other hand, in hot conditions, feed intake is significantly reduced, and it becomes impossible to provide the right amount of nutrients. The publication only mentions temperature, how were the animals held? Please insert text about husbandry conditions: lighting intensity and duration, gaseous ammonia, temperature, etc.

Author: We have clarified the temperature and light conditions in the material and methods section in lines 113-119. Unfortunately, NH3 or CO2 were not recorded. 

  1. Why were specifically the described genotypes chosen. Please discuss. (Different genotypes may also need different fatting lengths.)

Author: The genotypes used in the present trial were the ones that are used as per common practice in the pig research facility and are one of the most commons in commercial pig production. Nevertheless, we have discussed the fact that different genotypes may result in different outcomes in the discussion section in lines 406-407.

  1. General remarks.

Author: We addressed your corrections and changed them through the manuscript.

Round 2

Reviewer 1 Report

Thank you for following all the suggestions.

Best Regards